# Hydrophobicity and Macroscale Tribology Behavior of Stearic Acid/Hydroxypropyl Methylcellulose Dual-Layer Composite

**DOI:** 10.3390/ma14247707

**Published:** 2021-12-13

**Authors:** Shih-Chen Shi, Yao-Qing Peng

**Affiliations:** Department of Mechanical Engineering, National Cheng Kung University, Tainan 70101, Taiwan; n16041561@mail.ncku.edu.tw

**Keywords:** macroscale abrasion, third-body, particle shape, solid lubricant

## Abstract

Hydroxypropyl methylcellulose (HPMC) and stearic acid (SA) are integrated to fabricate a double-layer thin film composite material with potential applications in sustainable packaging and coating materials. The effect of SA concentration on the moisture and wear resistance at the macroscale of the composite are studied. The amount of SA on the surface (>SA5H) is beneficial in increasing anti-wear behavior and reducing the friction coefficient by 25%. The petal-shaped crystals formed by SA are distributed on the surface of the double-layer film, increasing its hydrophobicity. When subjected to wear, the SA crystals on the surface of the double-layer film are fractured into debris-like abrasive particles, forming an optimal third-body of moderate shape and particle size, and imparting anti-wear and lubricating characteristics.

## 1. Introduction

In the recent years, concerted efforts were applied to environmental protection owing to the rising awareness of environmental issues. In particular, the environmental effects of plastics are an extensively studied topic. The environmental detriment caused by synthetic plastics can be divided into two ways. First, most plastics are derived from petrochemicals, and the processes and products of the petrochemical industry are accompanied by a massive increase in carbon footprint. This increases the burden on the environment and intensifies the greenhouse effect [1]. Second, plastic waste cannot be completely decomposed by the environment. It forms plastic particles, which are discharged into the ocean through soil, groundwater, and lakes. These particles can then be ingested by fish, and enter the food chain. This causes ecological catastrophes and allows the accumulation of biological toxins, ultimately endangering human health [2]. Therefore, reducing environmental pollution caused by manufacturing, using, and discarding plastics has been at the forefront of several research initiatives.

According to the US Environmental Protection Agency, green engineering and green chemistry are two approaches to green tribology. Green engineering requires that when feasible in terms of technology and economics, all effort is concentrated in reducing the pollutants that are generated in the processes of product design, commercialization and use, and endanger the environment and human health. Green chemistry, also known as sustainable chemistry, requires reducing the use of chemical products, as well as the use of hazardous substances in manufacturing processes, while maximizing reaction efficiencies. In summary, the three core actions of green tribology entail using green materials that are environmentally-friendly and safe for humans; minimizing pollutants and harmful substances emitted during the design, manufacturing, and use of products; and maximizing lubrication and reaction efficiency [3]. Our research addresses all three core actions.

Cellulose is mostly derived from plants, and its extraction process can be performed via green chemistry [4]; it also meets the requirements of green engineering in application [5]. Hydroxypropyl methylcellulose (HPMC) is a chemical derivative of cellulose with a six-membered ring pyranose backbone, which forms an ether R-O-R structure with one another. In addition to its use in the outer coatings of food and medicine, its environmental-friendliness, nontoxicity to humans, biodegradability, and excellent mechanical properties have rendered it a commonly used material in green tribology [6,7]. The commonly used preparation processes of HPMC film are hot pressing [8], solvent evaporation [9], and 3D-printing method [10] in the recent years.

Stearic acid (SA) is a saturated fatty acid composed of a long chain of 18 carbon atoms at the hydrophobic end and a carboxyl group at the hydrophilic end. Hence, its amphiphilic nature enables its use as an active interface agent. Further, owing to its long carbon chain, it exhibits strong hydrophobicity and is used to protect hydrophilic materials that are susceptible to moisture [11]. Furthermore, the SA molecules can self-assemble to form micelles in an aqueous solution, causing them to form a turbid hydrocolloid [12]. The formation of micelles indicates that the SA molecules are agglomerated. However, because of the increase in volume and weight, precipitation is prone to occur, which prevents the SA molecules from being uniformly distributed in the aqueous solution. The results from a simulation study on the distribution of SA and HPMC molecules in HPMC aqueous solutions of different concentrations based on dissipative particle dynamics were compared to the experimental particle size distribution of SA micelles [13]. It was deduced that when the number of HPMC molecules (i.e., concentration) is greater than or equal to that of the SA molecules, the long-chained HPMC molecules trap the SA molecules and restrict their free motion, preventing the contact of SA molecules with the water molecules. Thus, small micelles are formed between the SA molecules, which prevent precipitation. In the SA/HPMC composite system, not only the hydrophobicity and lubricating properties of SA benefit HPMC, but HPMC acts as a colloid stabilizer such that SA molecules are evenly distributed in the mixed solution.

In addition to their hydrophobicity, SA molecules can also be adsorbed onto metals by the tribo-chemical reaction triggered by the wear process to form a molecular film with low shear strength, load-bearing capacity, and lubricating properties [14,15,16]. To further understand the lubricating mechanism of SA molecules, several hypotheses have been proposed to explain the adsorption process of SA molecules onto metals. The review by Kajdas concluded that the tribo-emission electrons and thermal electrons dissipated during the wear process may act as catalysts for the formation of carboxylates (ROOFe) between the SA molecules and the metal [17]. Loehlé et al. performed simulations from the perspective of molecular dynamics. The carboxyl end O-H group in the SA molecule has the weakest bonding energy (−14 kcal/mol); hence, it is most susceptible to breakage by the physical action of wear and causes hydrogen ions to dissociate. After dissociation, the SA molecules form carboxylates (ROO-), which bond with metal [18].

The ATR-FTIR signal on the metal surface after the wear process with SA molecules has a wave number of approximately 1460 cm^−1^ and 1600 cm^−1^, where symmetric and asymmetric stretching of carboxylates occurs. The spacing between the wave numbers of the characteristic peaks is less than 300 cm^−1^, and the characteristic peaks at 1460 cm^−1^ have intensities greater than those at 1600 cm^−1^ [17,18,19,20]. This indicates that the SA molecules and the metal mostly form bridging or bidentate, causing symmetric adsorption [20,21]. It is speculated that the bonding in symmetric adsorption is relatively stable, making desorption during physical or chemical effects during wear less likely [18].

Hagenmaier et al. added SA molecules to a HPMC solution and prepared a composite coating by evaporating the solvent. SA molecules at high concentrations form crystals on the surface of the coating, which can significantly improve its resistance to moisture [22]. Further, a composite coating was prepared by adding fatty acid molecules with different numbers of carbon atoms and degrees of saturation to a HPMC aqueous solution [23]. They found that the size of the colloid formed by the SA molecules in the aqueous solution is related to the microstructure of the composite film formed after evaporation, which also affects the surface roughness, transparency, mechanical properties, and water vapor permeability of the film. In addition, Fahs et al. revealed that in low-concentration SA/HPMC composite coatings, the SA molecules migrate to the HPMC surface, greatly reducing the surface roughness, capillary action, and surface energy of the film, while simultaneously lowering the microscopic and macroscopic coefficients of friction to 0.1 [24]. 

The HPMC film is environmentally friendly, exhibits good mechanical properties, gas barrier capacity, and lubricating properties with the potential to be used as a packaging and coating material. Furthermore, HPMC has self-healing properties owing to its hydrophilicity, which enable it to repair wear scars in the presence of solvents or moisture [25,26]. However, it is also susceptible to moisture, which limits its application. Hydrophobic additives such as SA [22] and amylose-sodium palmitate [27] were used to increase the hydrophobic property and reduce the water vapor permeability of pure HPMC. While improving the hydrophobicity of composite materials, the tribology behavior of the composite coating can be effectively enhanced [24]. The addition of MoS_2_ with lubricating behavior and high load capacity additives, such as Al_2_O_3_, CuO, Al, and Cu, have a positive benefit to improving HPMC matrix composite coatings’ wear resistance [5,28]. In this study, we apply hydrophobic SA molecules that have a lubricating effect with the hydrophilic HPMC to form an SA/HPMC composite dry coating, which can improve the moisture resistance of HPMC and enhance its macroscopic wear properties. Although there are several studies that discuss the wear properties and wear mechanism of SA molecules under wet wear and at the microscale, those on SA molecules under dry wear at the macroscale are limited, and dry wear conditions determine applicability for use as packaging material. Using surface structures to improve hydrophobic properties has been frequently researched with biomimetic structures, such as lotus leaves and rose petals, to improve the hydrophobicity of materials, reduce surface energy and adhesiveness, impart a waterproofing effect, and improve wear properties [29,30,31,32]. In addition, the use of bio-based materials for packaging or as a coating to improve waterproofing, corrosion resistance, and wear properties of paper to replace the use of polymers, such as polyethylene, has been investigated [33]. This study can enable us to determine the key factors of SA molecules that affect moisture resistance and wear properties of HPMC composites, which can then be used as a reference for design parameters of packaging material.

## 2. Materials and Methods

### 2.1. Preparation of Composite Film 

HPMC and SA solutions were separately prepared. First, 100 mL of deionized water was heated to 80 °C with an electromagnetic heating stirrer and 3 g of HPMC (USP2910, PHARMACOAT 606, Shin-Etsu, Tokyo, Japan) powder was added, followed by stirring the solution for 1 h, and storing it at 5 °C. The SA solution was prepared by adding 0.15 g, 0.3 g, 0.6 g, and 1.2 g of SA powder (99% purity, Pharma-Up Enterprise Co., Ltd., Taipei, Taiwan) to 100 mL of ethanol (95% concentration, Echo Chemical, Miaoli, Taiwan); the solution was stirred at 25 °C for 1 h and stored at 25 °C. Then, 300 μL of the HPMC aqueous solution was injected by a micropipette on a silicon substrate and placed in an oven maintained at 30 °C and relative humidity (RH) of 40% for six hours to form the HPMC coating. Micropipette injection of 100 μL SA solution with five different concentrations (i.e., 2.5 mM (SA2.5H), 5 mM (SA5H), 10 mM (SA10H), 20 mM (SA20H), and 40 mM (SA40H)) onto the pre-made HPMC coatings was prepared and placed in a controlled environment (30 °C, RH 40%) for 1 h. Then, the prepared dual-layer film was transferred and stored in a temperature control moisture-proof container (25 °C, RH 30%).

### 2.2. Analysis of Stearic/HPMC Dual-Layer Film

The morphology of the composite coating film was examined using scanning electron microscopy (SEM, XL-40FEG, Philips, Amsterdam, The Netherlands) at 15 kV and a working distance of 15 mm. The chemical structure of the composite surface was monitored using attenuated total reflectance Fourier-transform infrared spectroscopy (ATR-FTIR, Thermo Nicolet NEXUS 470, GMI, Golden Valley, MN, USA). The characteristics of the coating, such as the surface profile, film thickness, and wear volume, were recorded using a 3D laser profiler (VK9700, Keyence, Osaka, Japan). The water contact angle of the coating was measured using a contact angle meter (FTA-1000B, First Ten Angstroms, Portsmouth, UK).

### 2.3. Lab Scale Friction Analytics

The tribological behavior of the composite was evaluated using a ball-on-disk tribometer (POD–FM406–10NT, Fu Li Fong Precision Machine, Kaohsiung, Taiwan) under a loading and disk speed of 0.03 m/s. The short-range wear test was used to observe the tribological behavior of various coatings under varying loads (2 N, 5 N, and 8 N) with a sliding distance of 1 m. The long-distance wear test was used to observe the tribology mechanism of the composite coating, where the distance was varied from 1 m to 10 m to 30 m, and the load was fixed at 2 N. A chrome steel ball (52,100 steel) with a diameter of 6.31 mm was employed as the upper counter ball, and the composite film was used as the lower disk test piece. The wear test was performed at 25 °C and RH 70%. The friction coefficient of the coating was monitored and recorded in real time, and the wear depth was measured using a 3D laser scanning microscope. The testing of each parameter was repeated three times.

## 3. Results and Discussion

### 3.1. Characterization of SA/HPMC Dual-Layer Film

The surface morphology of the double-layer composite material is shown in Figure 1a–f. The SA molecules form petal-shaped crystals that are unevenly distributed on the HPMC. With increasing SA concentration, the distribution density of the SA crystals on the surface of the HPMC films also increased. In particular, the surface morphology of the SA40H double-layer film is consistent with the results obtained by Zhang et al., who coated cellulose with SA without a hot-pressing treatment [34]. The film thickness and surface roughness (Rz) of composites are shown in Figure 1g. The double-layer coating is prepared by HPMC film first, and then the SA crystallization layer. The thickness of the HPMC layer is controlled at 30 ± 2 μm. The SA crystal layer increases Rz significantly as the SA increases. 

The ATR-FTIR spectra of the SA5H film (Figure 2), with a relatively lower concentration of SA, is similar to that of the pure HPMC film, confirming the extremely low SA concentration on its surface. As the concentration of SA gradually increased, the signals of the long carbon chain of SA were first observed at 2910 cm^−1^ and 2850 cm^−1^. The ATR-FTIR spectra of SA20H is similar to that of pure SA, except for the signals that appear because of the C-O-C structure of HPMC at 947 cm^−1^ and 1059 cm^−1^. The water vapor peaks at 1650 cm^−1^ and 3450 cm^−1^ are also reduced, as increasing the concentration of SA improved the hydrophobicity of the material surface. In addition, the amount of SA distributed on the surface of the SA/HPMC double-layer film also increased accordingly.

The water contact angle increased from 61° (HPMC) to 110° (SA40H) with increasing SA concentration as shown in Figure 3a. In Figure 1, the surface of the double-layer film is covered with SA crystals. As the amount of SA on the surface increased, the enhanced hydrophobicity caused the water contact angle to increase. These results corroborate with those by Zhang et al., who covered cellulose with SA crystals, and Garoff, who used a self-assembled monolayer of SA atomic film on cellulose [34,35]. The water contact angle changes with the centerline averaged roughness (Ra) value because of the change in the wettability model between the water droplet and the surface. Surfaces that can be described by the Cassie–Baxter model are often relatively rough. These surfaces form a large number of air pockets owing to the air trapped in the asperity valleys of relatively large volumes. In the water contact angle test, the contact interface changes from the original solid-liquid interface to the solid-liquid-gas three-phase contact. As a result, the solid-liquid contact area decreases, which ultimately leads to an increase in the water contact angle [36].

To further clarify the relationship between the geometric properties and wettability of the surface, Kubiak et al. also calculated the covariance coefficient of several two-dimensional parameters describing the surface and water contact angle. The relative material ratio (R_mr_) had the largest covariance coefficient (148.3); thus, it is most relevant to surface wettability among all parameters [37]. Therefore, Kubiak et al. assumed that the valleys below 1/4 of the maximum height of the surface roughness curve were all filled with air pockets, and estimated the solid-liquid and solid-gas contact area ratios based on the ratio of 1/4 the maximum height of the roughness curve to the material. The ratios were corrected using the Wenzel and Cassie–Baxter models, and it was found that the surfaces of the same material with different geometric properties had approximately the same ideal water contact angle. This correction formula for the water contact angle is only applicable to two-dimensional surfaces with similar contours of wear scars in the vertical direction after polishing. Nevertheless, the assumption that the rough valleys below 1/4 of the maximum height of the roughness curve are filled with air pockets is also applicable to our double-layer film’s three-dimensional structure, especially because its surface is covered with petal-like SA crystals [37]. Figure 3b reveals the R_mr_ of the double-layer composite material, which can be derived as the ratio of the material length of the bearing area curve (BAC) at a specific height [38]. The ratio of the 1/4 maximum height of SA20H relative to the material is 85%, implying that the rough valleys of this structure are relatively large. This is consistent with the conditions described by the Cassie-Baxter model. The double-layer membrane composite material may be simultaneously affected by some of its surface geometric properties and the hydrophobic characteristics of the SA material. This results in a sharp increase in the water contact angle and a hydrophobic surface is finally obtained. The load curve is obtained by adding the curve length corresponding to each height, which is used to describe the ability of the curve to bear a load [38]. The load curve shows (Figure 3b) that the curve ratio of the double-layer film at a relatively high position was comparatively small. This indicates that without considering the plastic deformation, it will bear the load with a smaller number of rough peaks, i.e., a smaller contact area [39] when the double-layer film is pressed to the same depth. According to Briscoe et al., the wear phenomenon of polymer films conforms to the cold wielding theory proposed by Tabor and Bowden, which is related to the shearing strength of the material and the actual contact area between the parts under wear. When the actual contact area of the parts under wear is small, a smaller coefficient of friction results, causing less adhesion wear [40].

### 3.2. Tribology Behavior of SA/HPMC Dual-Layer Film

Figure 4a shows the average coefficient of friction and wear scar depth of the SA/HPMC composite coating under 2 N, 5 N, and 8 N loading at a sliding distance of 1 m. As the concentration of SA increased, the coefficient of friction remained relatively constant, while the depth of wear scars decreased. On the contrary, when the load increased, the coefficient of friction of the coating with the same SA concentration decreased, while the wear scar depth increased. Under a 2 N load, the coefficient of friction of SA2.5H is 0.14, which is significantly lower than that of pure HPMC (0.51). This may be attributed to the lower surface energy of SA molecules on the HPMC surface [24]; here, the influence of surface roughness of the double layer film is ignored for purposes of simplicity. 

The coefficients of friction of the films with the same process parameters under different loads were compared. Under 5 N and 8 N loads, the coefficient of friction is maintained between 0.10 and 0.12, which is significantly smaller than when the load is 2 N (Figure 4a). This shows that SA can provide a better lubrication effect under loads of 5 N and 8 N. These results imply that at the sliding distance of 1 m, the SA petal-shaped crystals on the surface of the SA/HPMC double-layer film are flattened, crushed, and even removed by the hard surface of the metal, which can drastically change the surface profile. However, the wear scar morphology (Figure 4b–d) of the double-layer film suggests that HPMC is protected by the remaining SA crystals, implying that only a small part of the composite is in direct contact with the metal surface and only limited quantities are removed.

Figure 4b–d show the wear scar morphology of the double-layer film with different SA concentrations at a sliding distance of 1 m under a 2 N load. Figure 4e–g shows the wear scar morphology of SA20H with loads of 2 N, 5 N, and 8 N at a sliding distance of 1 m. The wear scar morphology of the double-layer film with low SA concentration is similar to that of films with high SA concentrations; the latter contain the debris formed by the scattering of the SA crystals under load and shearing force [6]. Regardless of the SA concentration, adhesion/wear or scratches in the wear scars were not apparent. 

Figure 5 shows the edge of the wear scars in SA20H, which contains a relatively high SA concentration. It can be observed that the SA petal-shaped crystals were not completely fractured, and the morphological structure of the SA20H surface under load was similar to that of the wear scar

As the wearing process progressed, SA produced a transfer layer on the specimen as shown in Figure 6. The petal-shaped SA crystals have not been completely squashed, and their surface under load is similar to the internal morphology of the wear scar. During the wear process, it was clearly observed that the SA crystals were squashed, indicating that in the early stage of wear (short-range wear), the SA crystals distributed on the surface will directly bear the load and deform.

During the long-distance wear test, the coefficient of friction of the SA/HPMC double-layer film remained low and stable over the entire 30-m sliding distance as shown in Figure 7. This phenomenon indicates that within a sliding distance of 30 m, the tribological substance in the interface can provide stable wear resistance. In particular, the COF of SA5H shows significant fluctuations. In addition, cracking and delamination occur at the wear scar of the test piece after wear, which is speculated to be caused by the test piece being continuously subjected to wearing at the unbroken and broken locations during rotation. Compared to the pure HPMC and SA40H, the COF effectively decreases from 0.2 to 0.15, which indicates that the existence of SA provides a low and stable tribology circumstance [41].

Figure 8a shows the wear depth of the double-layer film at different distances under a load of 2 N. The rate of increase of the wear depth of all the double-layer films at a unit sliding distance of 30 m is significantly slower than at 10 m. Moreover, the wear of the double-layer film did not decrease, but increased with increasing SA concentration. The measured change in the wear scar depth of the double-layer film is indicative of the removal rate of SA, that is, the petal-shaped SA crystals on the film surface are damaged, but some remain adhered to the metal surface in contact with the abrasive parts, as discussed above. Furthermore, the higher the concentration of the SA crystals on the HPMC surface, the larger the crystal volume; thus, the crystals are more susceptible to damage and removal. Therefore, the wear tends to increase with increasing SA crystal concentration. 

The tendency of the increased rate of the wear depth per unit sliding distance is further analyzed to demonstrate that it decreases as the sliding distance increases. Figure 8b–d shows the wear scar surface morphology of the composite coating at higher magnifications at different sliding distances (1 m, 10 m, and 30 m, respectively). The significant difference lies in the appearance of SA. Based on the analysis of the short-range wear test, the SA debris in the 1 m wear scar is mainly due to the petal-shaped crystals being under load. However, at 10 m and 30 m, the SA wear debris appears to be shredded. It is speculated that during the wear process, the relatively large crystals at 1 m will continue to be subjected to wear stress and become rod-shaped at 10 m, and this morphology will not significantly change until 30 m. In the early stages of wear, SA is squeezed, squashed, and broken (marked A in Figure 8b). These flattened SA crystals exist in a needle-like form under continuous load. In the later stage, as wear continues to occur, the needle-like rods are rolled into finer forms (Figure 8c, mark B and Figure 8d, mark C). When the SA in the wear scars is in the needle form, the double-layer film has a much lower material removal rate than bulk crystals, which is a relatively ideal grain morphology. According to the third body theory, needle-like SA lubricating particles have an additional velocity accommodation mode of rolling compared to petal-shaped crystals. Moreover, it is difficult for smaller size particles to move away from the wear contact area; thus, the needle-like SA particles continue to work during the wear process without being easily removed [42,43].

Figure 9 shows the wear mechanism of the SA/HPMC double-layer composite film. Although the large SA crystals provide lubrication and protect the HPMC substrate material, they are also easily removed from the contact area. In contrast, when the SA crystals are rolled into needles, the material removal rate decreases. Therefore, when designing SA/HPMC composite coatings that require enhanced wear properties, it is not necessary to increase the SA content on the surface, but the amount of SA crystals distributed in the wear scars is sufficient to produce the ideal shape of the abrasive particles during the wear process.

## 4. Conclusions

In this study, the effect of SA on the moisture resistance and wear properties of an SA/HPMC double-layer composite was investigated. SEM studies showed that the SA molecules form crystals and are distributed on the surface of the double-layer film. As the concentration of SA solution increased, the thickness of the coating slightly increased, while the amount of distributed SA significantly increased. In the case of SA10H, SA20H, and SA40H, SA forms a petal-shaped crystal. Furthermore, increasing the amount of SA crystals distributed on the surface of HPMC increased the surface roughness and water contact angle of the double-layer film, which increased the load-bearing capacity and hydrophobicity of the coating, respectively. The wear test results showed that the double-layer composite material demonstrates a higher coefficient of friction and better wear life than those of the pure HPMC film, indicating that the addition of SA can effectively improve the wear properties of HPMC at the macroscale. The wear mechanism of the double-layer film involves the formation of a new interface of the abraded crystalline debris between the HPMC and the metal, providing lubrication and protection for the HPMC. When SA changes from large crystals to the needle form, the coating has a lower SA removal rate. This is presumably because the needle-like third-body layer provides a velocity accommodation mode to the rolling speed of the SA molecules, and their smaller size makes disengaging from the contact area in the third-body flow less likely. 

Our results indicate that a hydrophilic surface, such as HPMC, can be tailored for desirable properties by modifying it with hydrophilic materials, such as SA. When hydrophobicity of the film surface is required, SA should form a special structure sufficient to fill a large number of air pockets in the rough valleys and be distributed on the hydrophilic material. In this case, the higher the SA content, the more hydrophobic the material, such as SA20H and SA40H. However, when wearing properties, such as a low coefficient of friction or wear resistance, are desirable and distribute sufficient SA, it is essential to ensure that it is distributed in the optimal third-body shape in the wear scars to provide continuous lubrication during the wear process. This is because the large SA crystals distributed on the surface of the material are easily removed. For example, SA5H has an excellent coefficient of friction and slight wear, which is ideal for anti-wear properties. 

This study successfully created composites from two natural green chemistry-compliant materials and demonstrated their green tribology properties, which can be used as a reference for hydrophobic packaging material design. However, this article does not discuss the behavior of composite materials in high humidity environments or even in direct contact with water. In the future, research will be conducted on how to supplement SA composite materials consumed in wear and their tribology behavior in high humidity environments.

## Figures and Tables

**Figure 1 materials-14-07707-f001:**
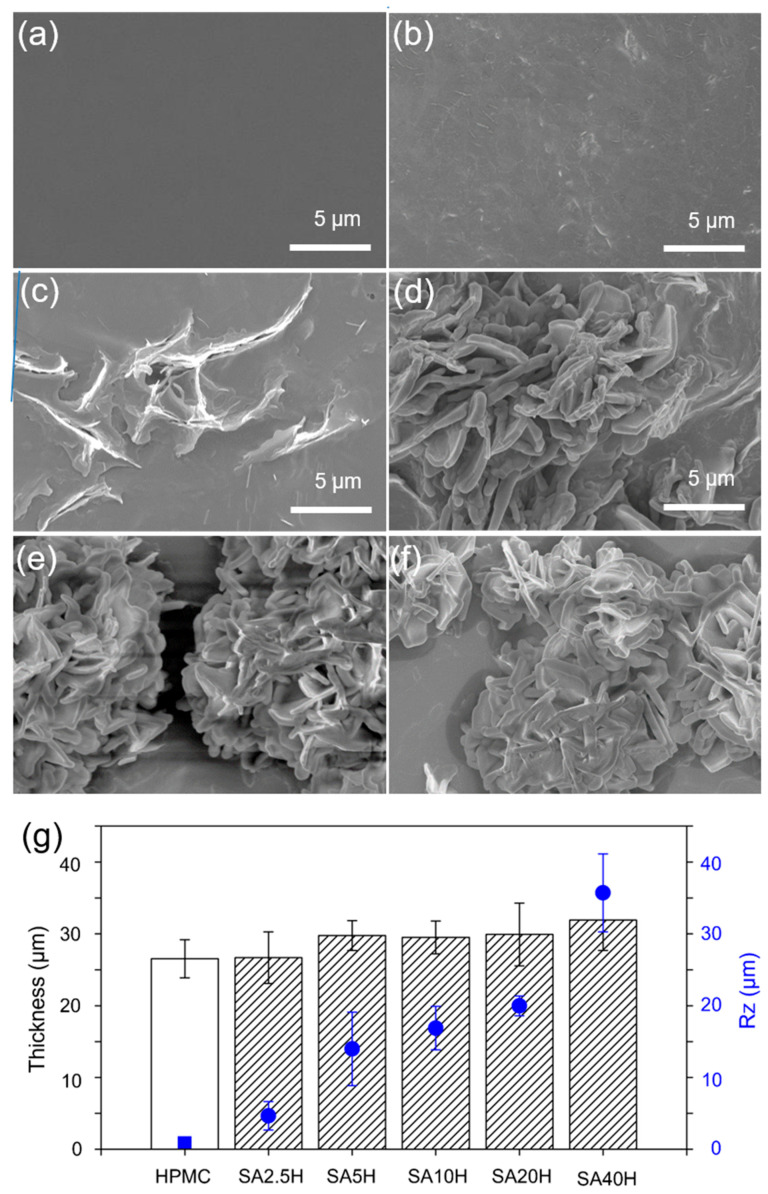
Surface morphology of (**a**) pure HPMC, (**b**) SA2.5H, (**c**) SA5H, (**d**) SA10H, (**e**) SA20H, (**f**) SA40H, and (**g**) film thickness and surface roughness (Rz) of composites.

**Figure 2 materials-14-07707-f002:**
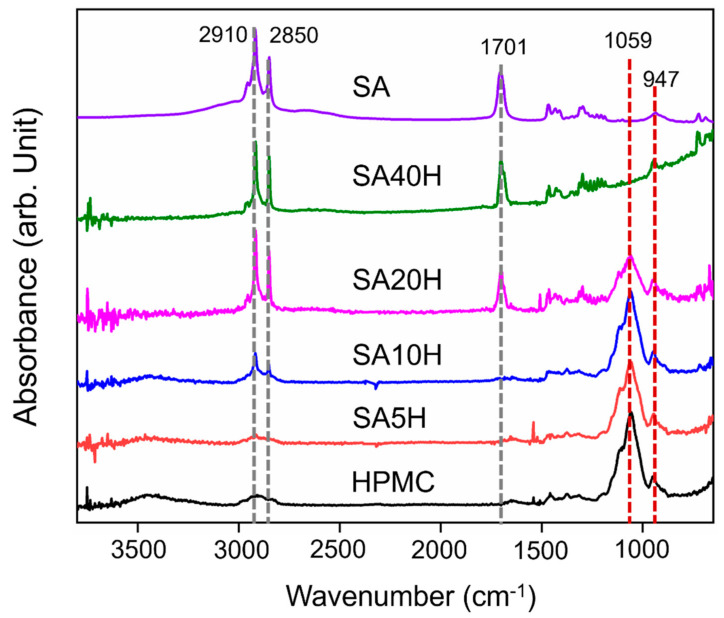
ATR-FTIR spectrum analysis of composite films.

**Figure 3 materials-14-07707-f003:**
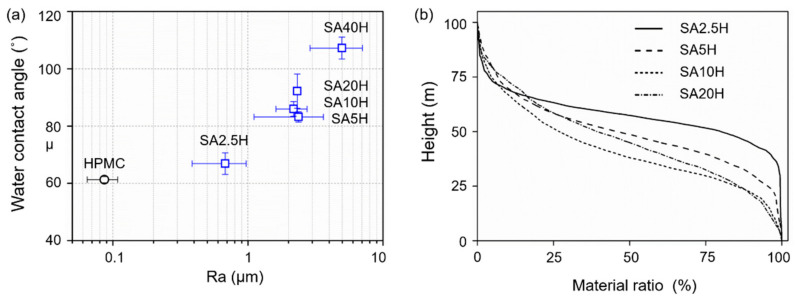
(**a**) Water contact angle and surface roughness, and (**b**) bearing area curve (BAC) of the dual-layer composites.

**Figure 4 materials-14-07707-f004:**
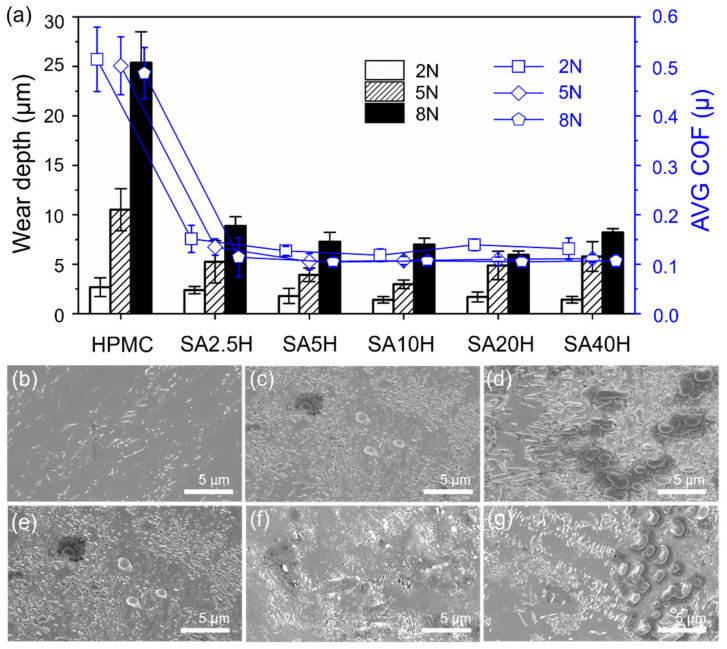
(**a**) Short-range wear depth and average coefficient of friction. Interface observation of (**b**) SA5H, (**c**) SA20H, and (**d**) SA40H. The SA20H surface under (**e**) 2 N, (**f**) 5 N, and (**g**) 8 N.

**Figure 5 materials-14-07707-f005:**
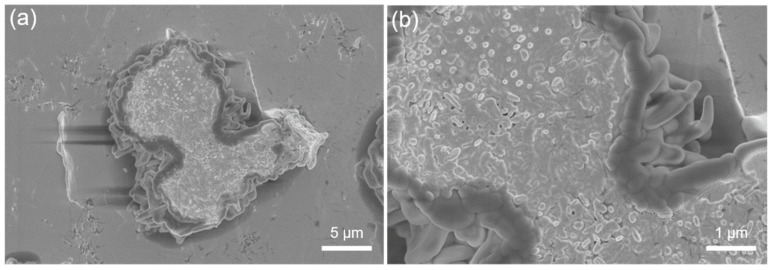
Surface morphology of wear scar edge of SA20H, (**a**) low magnification, (**b**) high magnification.

**Figure 6 materials-14-07707-f006:**
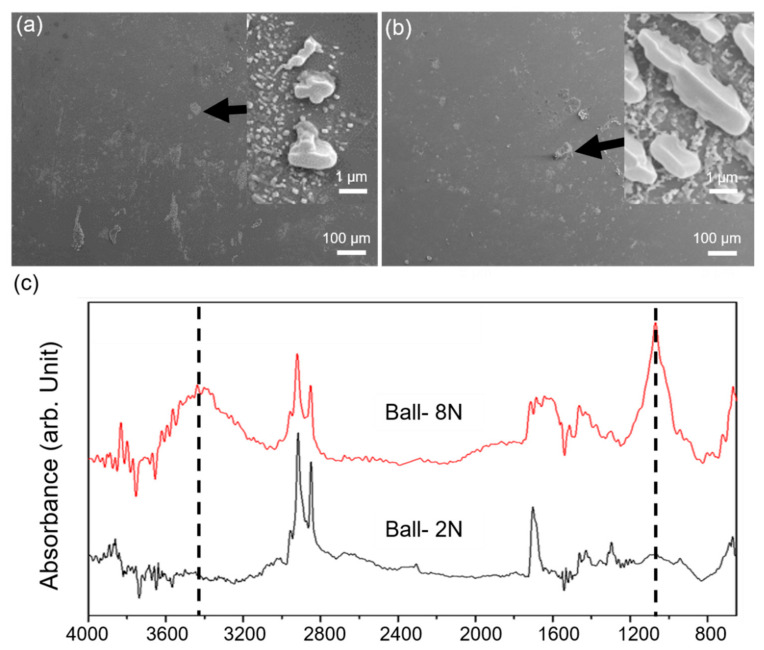
Surface observation of upper counter ball, (**a**) 2 N, (**b**) 8 N, (**c**) ATR-FTIR analysis on the counter ball surface (SA20H, 1 m).

**Figure 7 materials-14-07707-f007:**
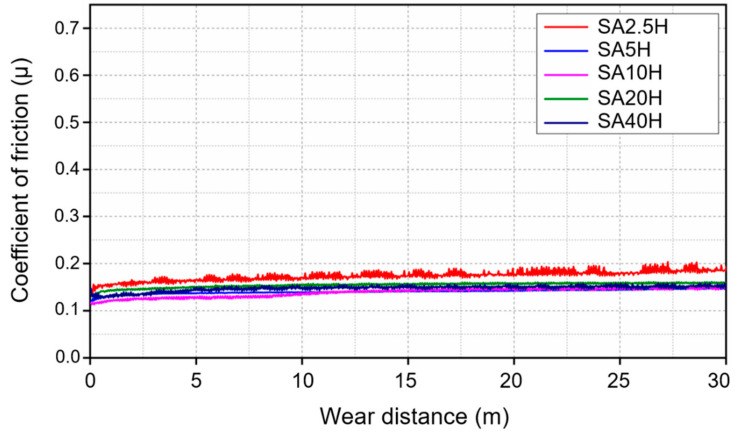
Coefficients of friction for coatings under long-distance wear (SA20H, 30 m).

**Figure 8 materials-14-07707-f008:**
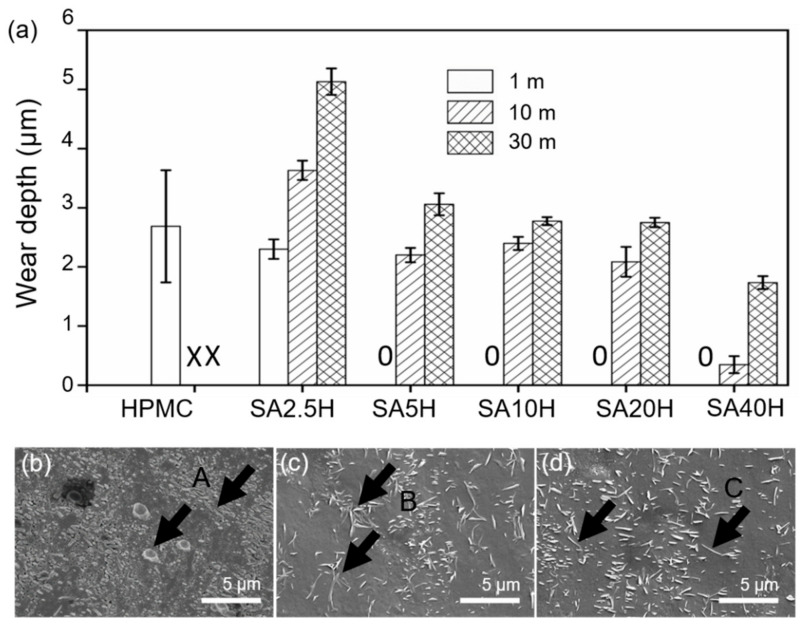
(**a**) Wear performance of composite coating. Wear depth and surface morphology at (**b**) 1 m, (**c**) 10 m, and (**d**) 30 m (SA20H, 2 N; X represents that the film is worn through; 0 represents that the wear depth is 0).

**Figure 9 materials-14-07707-f009:**
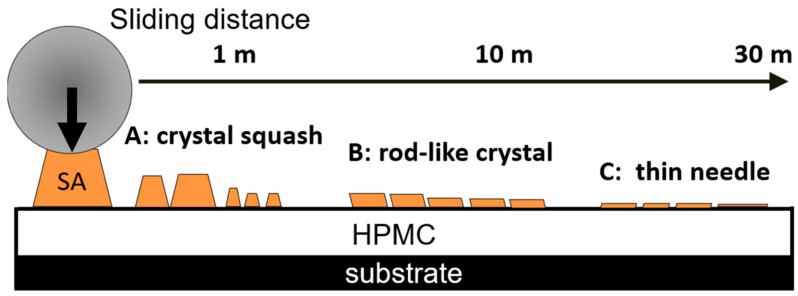
Schematic diagram of wear mechanism.

## Data Availability

The data presented in this study are available upon request from the corresponding author.

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
