# Peer review of "Hydrophobicity and Macroscale Tribology Behavior of Stearic Acid/Hydroxypropyl Methylcellulose Dual-Layer Composite"

_materials, 2021, doi:10.3390/ma14247707_

Round 1
Reviewer 1 Report
The manuscript “Hydrophobicity and macroscale tribology behavior of Steric acid/cellulose dual-layer composite” deals with a dual layer composite material made of hydroxypropyl methylcellulose and stearic acid with potential applications in packaging and coating materials.
The manuscript can be published after some revisions:
Title : steric acid: wrong spelling and why cellulose instead of hydroxypropyl methylcellulose?
Abstract: I think that in the abstract results obtained can be more detailed like in the first part of the conclusion. In this study ………………..flow less likely.
Page 4 line 1: FTIR was use in Attenuated Total Reflectance (ATR) mode? Add the technique for experimental reproducibility.
Figure 1. Please add in the Figure 1 the surface morphology of HPMC and SA2.5H films. How many replications were tested? Different films or different measure in the same film? Add it in the experimental part.
Figure 4. How many replications were tested? Different experiments or different replications in the same experiment? Add it in the experimental part.
Reviewer 2 Report
Shi and Peng's manuscript describes the synthesis and characterization of stearic acid (SA), and hydroxypropyl methylcellulose (HPMC) integrated double-layer thin film composite material. Further, moisture and wear resistance studies of the synthesized thin-film composite material were also done. The current study may be helpful for future applications of sustainable packaging and coating materials. The authors have given a detailed overview of the topic and justified the research question. The study method is valid and reliable. In conclusion, the manuscript may consider publication in this journal with some corrections and suggestions that I think improve the paper.
- Page 3; 2.1; Preparation of composite film: Additional details about conditions of preparation of composite film should be given. For example, the conditions of SA coating are not provided. Further, the applied procedure for hot-pressing to make each of these coatings should be given.
- Page 5; Figure 1: Image of surface morphology of HPMC should be given.
- Page 5; Figure 2: IR of only SA on the top of the figure will be helpful to compare the results.
Reviewer 3 Report
Manuscript entitled “Hydrophobicity and macroscale tribology behavior of Steric acid/cellulose dual-layer composite”
The presented manuscript is well written and organized in a good manner. The characterization techniques which were used are appropriate. Using cellulose derivative hydroxypropyl methyl cellulose in a combination with steric acid is a promising composite for use in e.g. food packaging due to good hydrophobicity and wear resistance. However, there are some issues that authors need to clarify, before reconsidering for publication. The biggest concern in my opinion is the chosen methodology for layers coating.
Here are my comments:
- There are various existing methods for thin films production and deposition. Could you please, in introduction part, briefly describe methods of production of HPMC films and their relation to wear resistance? Is there some method which is superior to others?
- Can you provide readers the information what are other additives that can be used in production of HPMC films for obtaining hydrophobicity and/or wear resistance?
- Methodology of film coating is somewhat problematic- I am a little confused by so called “smeared“ coating? Why you used smearing? With what you smeared solution of HPMC or SA? It seems that applied technique is not reproducible? If you used 300 μl for coating what is the surface area of the sample? How did you control the thickness of layer? Please provide detailed description and if possible illustration of the method.
- How was it possible to control the thickness and evenness of the HPMC layer at 30±2 μm by smearing technique?
- Why HPMC aqueous solution was stored at 5ºC? Why did you choose 30 ºC for drying? Is it enough to remove all water and moisture from the layer? After adding HPMC and then SA, how and where samples were stored for further characterization? When SA was coated onto HPMC, were they dried in an oven or in the open air, and for how long?
- Was HPMC with SA after drying removed from silicon substrate? If so, please provide SEM image of another side of the film, where HPMC is?
- How many samples were prepared? You expressed thickness of samples with deviation of 2 μm, how many samples were tested? Please provide information for all techniques (wettability, wear resistance).
In my opinion, authors need to address and clarify these methodology issues, before reconsidering the manuscript for publication.
Round 2
Reviewer 3 Report
The authors have properly addressed all concerns and explained their methodology in detail. It was a pleasure reading the revised version of the manuscript, and I recommend this article to be published. In answer to the review, the scheme of coatings preparation is very clear and nicely presented, therefore I recommend authors, if they wish, to either include it in the manuscript or in the graphical abstract.